# TRANSALUD: A qualitative study of the healthcare experiences of transgender people in Barcelona (Spain)

Kevin Santander-Morillas[1,2], Juan M. Leyva-Moral[3]*, Marta Villar-Salgueiro[4], Mariela Aguayo-González[3], David Téllez-Velasco[1,2], Nina Granel-Giménez[3], Rebeca Gómez-Ibáñez[3]

1 Hospital Vall d'Hebron, Barcelona, Spain, 2 Nursing Department, Faculty of Medicine, Universitat Autònoma de Barcelona, Spain, 3 Nursing Research Group in Vulnerability and Health (GRIVIS), Nursing Department, Faculty of Medicine, Universitat Autònoma de Barcelona, Barcelona, Spain, 4 Stop NGO, Barcelona, Spain

* JuanManuel.Leyva@uab.cat

**Data Availability Statement:** Data cannot be shared publicly because of ethical restrictions on interview transcripts. Data are available from the

## Abstract

Transgender identities are still considered a psychiatric pathology in many countries according to the prevailing biomedical model. However, in recent years, this pathologizing vision has begun to shift towards a perspective that focuses on the diversity of transgender peoples' experiences. However, some transgender people still face denial of services, discrimination, harassment, and even violence by healthcare professionals, causing them to avoid seeking ongoing or preventive healthcare. This article describes the health experiences of transgender people in Barcelona regarding their access and use of non-specialized health services. Semi-structured interviews were conducted using a descriptive phenomenological approach with sixteen transgender people between December 2018 and July 2019. The data were analyzed descriptively and thematically following the method proposed by Colaizzi with the help of the Atlas.ti8 software. Transgender people care experiences were divided into three categories: overcoming obstacles, training queries, and coping strategies. Participants identified negative experiences and difficulties with the health system due to healthcare providers' lack of competence. Discriminatory, authoritarian, and paternalist behaviors are still present and hinder the therapeutic relationship, care, and access to healthcare services. There is a fundamental need for the depathologization of transgender reality and training for healthcare professionals in the field of sexual diversity. Training in sexual and gender diversity must be included in the curricula of university courses in the health sciences.

## Introduction

Sex is legally assigned at birth as either male or female depending on the physical features of the newborn. However, it is possible that one's assigned sex does not match their gender identity, that is, one's own perception or feelings regarding one's gender [1]. This is the case of

Universitat Autònoma de Barcelona Ethical Committee (contact Dr. Jose Luis Molina via ceeah@uab.cat) for researchers who meet the criteria for access to confidential data.

**Funding:** The author(s) received no specific funding for this work.

**Competing interests:** The authors have declared that no competing interests exist.

transgender people, a vulnerable and little-visible group today [2]. Transgender is an umbrella concept that is both general and inclusive and encompasses those people who identify with a gender different from the one they were assigned at birth or who express their gender identity in a non-normative way in relation to their biological sex [3]. This concept encompasses many different gender identities and expressions: trans, genderqueer, transsexual, gender fluid, or third gender, among others [4].

Currently, the transgender condition continues to be pathologized through the diagnosis of gender dysphoria as of the fifth version of the *Diagnostic and Statistical Manual of Mental Disorders* for psychiatric diagnosis published in 2013 [5, 6]. However, authors such as Coll-Planas and Missé [7] criticize this pathologization, stating that when people do not adapt to the norms established by society (concordance between their assigned gender and their own gender identification, or identification within the male-female binary at all), their mental health is questioned instead of understanding and accepting the diversity and complex reality of each individual.

At the legislative level, the Organic Law 3/2007 of March 15 in Spain stipulated the medical diagnosis of gender dysphoria and a minimum hormonal treatment of two years as requirements for the modification of sex in official documents [8]. At the end of 2016, as a result of the Catalan Law 11/2014 in favor of LGBTI rights and the struggle of different associations of transgender people, the Catalan Department of Health launched a new model of healthcare for transgender people, which was approved the following year and implemented in November 2017 [9]. This model proposes the concept of the transgender condition as a diverse gender expression and not as a disease, with the aim that transgender people do not have to be psychiatrically diagnosed in order to access the treatments they require and thus to provide them with comprehensive healthcare.

The Institute of Medicine's 2011 *Report Lesbian, Gay, Bisexual, and Transgender Health: Building a Foundation for a Better Understanding* [10] describes how LGBT people are often marginalized within medical settings and are at increased risk of health disparities. Despite the increasing priority on realizing shared decision-making, little attention has been paid to the needs of LGBTIQ+ people in the healthcare setting [11].

Increasingly, care based on a humanistic model centered on the patient from a holistic view, is being promoted among healthcare professionals. However, the vast majority do not possess sufficient competencies to offer transgender people comprehensive healthcare that can respond to their needs and concerns, causing a deficit in the provision of health services [12, 13]. This is mostly due to a theoretical ignorance produced by a deficit of specific training in transgender issues, as well as a scarce scientific production and research and little promotion of comprehensive LGBTIQ+ health policies [14]. This was reflected a few years ago in a study by Whittle, Turner, Combs, and Rhodes that found that 30% of transgender people had experienced a situation in which a health professional wanted to help them but lacked information on how to do so [15]. More broadly, subsequent research has detailed a high percentage of healthcare professionals (73%) who believe they have not received sufficient university training in sexual health, which creates a significant barrier during healthcare practice [16, 17].

In addition, some transgender people face denial of services, discrimination, harassment, and even violence by healthcare professionals, causing them to avoid seeking ongoing or preventive healthcare [4, 18]. According to European data, 22% of transgender people who seek health services feel that health personnel treat them in a discriminatory manner because they are transgender (20% in Spain), and 30% report negative experiences in the care they receive from health professionals [19]. According to the latest FELGTB report for the Spanish territory, 48% of the transgender people surveyed had ever felt discriminatory or inadequate treatment by healthcare personnel [20]. Scientific evidence shows the presence of challenges during

interactions with health systems that make it difficult to develop and obtain specific care for transgender people [21, 22] in short, discriminatory treatment is evident. Rodriguez et al. [23] identified a statistically significant correlation between being recognized as a transgender person and experiencing discrimination in healthcare settings.

However, there is a lack of qualitative evidence in this regard. Consequently, the aim of this study is to describe the experiences of transgender people in relation to the healthcare they receive in primary care and hospital services in Barcelona in the period 2017–2019. The results will help to understand their experiences, make them visible, and identify areas for improvement.

## Methodology

This was a descriptive qualitative study using a phenomenological approach [24]. Data were collected through semi-structured interviews in person and by telephone, according to the preference of the participants. Transgender people over 18 years of age who were residents in Barcelona and who had attended a hospital or primary care center in Barcelona in the last two years were invited to participate. Community organizations and health centers specializing in the care of transgender people participated in the dissemination of the study and recruitment. Snowball sampling was performed. The intention was to interview ten participants, although the final sample size was defined by the saturation of the data [25].

The interview script (see Table 1) was agreed upon by two experts, and a pilot test was conducted with a participant whose data were not included in the study. Prior to the start of the interviews, the participants were informed of the objectives of the study and all its details. A space was offered to resolve possible doubts, and once all the questions had been answered, an informed consent form was signed. The interviews were conducted between December 2018 and July 2019 by three members of the research team, who agreed on how to proceed with the interview and how to resolve possible incidences. The duration of the interviews was variable, with an average of 30 minutes. All interviews were audio-recorded and transcribed verbatim after completion.

The data were analyzed descriptively and thematically following the method proposed by Colaizzi [26] with the help of the Atlas.ti8 software. The analysis was led by one of the researchers in constant interaction with the principal investigator, discussing the findings obtained at all times. The analytical procedure involved reading and rereading the transcripts to make sense of the participants' accounts, extract meaningful statements for the study phenomenon, formulate hidden meanings in the various contexts of the phenomenon studied, categorize the meanings into groups of themes common to all participants by validating the researchers' emerging conclusions with the participants' original stories, formulate a comprehensive

**Table 1. Sample questions from the semi-structured interviews.**

| |
|---|
| 1. How would you describe your experience with healthcare professionals? How did you feel you were treated? |
| 2. Tell me about the communication and relational skills of health and non-health professionals. How would you describe them? |
| 3. How would you describe the degree of knowledge of transgender issues among healthcare professionals? |
| 4. What is transphobia for you? Have you ever experienced it in a healthcare center? What impact did it have on your health? |
| 5. How would you describe the behaviors and attitudes of healthcare professionals? Has it had an impact on your life? Tell me about it. |
| 6. What worries/bothers you the most about healthcare professionals? Why? |
| 7. How do you think we can eliminate the negative experiences you have experienced? |

description integrating the meanings resulting from the categorized themes, generate a prototype theoretical model, validate the findings by returning them to the experts, and incorporate any changes in the final description [26, 27].

The study was approved by the ethical committee of the Universitat Autònoma de Barcelona (Spain). No data that could reveal the identity of the participants were collected, since pseudonyms were used in the transcripts and only the research team had access to the transcripts. The audio clips were deleted after they were transcribed. To ensure the credibility and consistency of the study, a detailed description of the data collection was presented, and the results were documented with quotations from the transcripts. The findings were contrasted and verified after several consensus meetings of the research team and the participation of two experts in qualitative research and gender identity issues. This study followed the Consolidated criteria for Reporting Qualitative research (COREQ) guidelines [28].

## Results

A total of 16 transgender persons participated, and their main characteristics are summarized in Table 2. Three categories were identified that describe the study phenomenon: 1) overcoming obstacles, 2) training queries, and 3) coping strategies. Caregiving experiences are described as a constant interplay between the difficulties and strengths that the respondents experienced. During this process, transgender people build their personal formulas to resolve situations in a resolute manner.

**Table 2. Participants' socio-demographic characteristics.**

| | |
|---|---|
| Age (mean ± SD) | 36.68 (13.24) |
| Gender Identity | |
| Trans woman | 11 |
| Trans man | 5 |
| Place of birth | |
| Spain | 9 |
| Other | 7 |
| Civil Status | |
| Single | 7 |
| Co-habiting | 6 |
| Divorced | 1 |
| Married | 1 |
| Other | 1 |
| Studies finished | |
| Primary | 1 |
| Secondary | 12 |
| College | 3 |
| Employment | |
| Employed | 12 |
| Unemployed | 4 |
| Financial Status | |
| Very low | 1 |
| Low | 6 |
| Medium | 5 |
| High | 4 |

## Overcoming obstacles

The care experiences of transgender people are mainly described as an accumulation of difficult experiences that they must spontaneously learn to manage. The majority identified having lived negative experiences in their contact with the health system with some professionals, both health and non-health professionals, describing these encounters as alienating, dehumanizing, and/or stigmatizing. Likewise, these experiences are closely related to the barriers identified in the access and use of health services and health professionals' lack of knowledge.

> A nurse was looking at me. . . She even got nervous when she saw me, the way I was talking. . . In this sense, I felt like. . . I did not feel well-treated in that sense. Because a professional person should know how to treat you, whether you are a woman or a man. . . I mean, he should be a professional (Participant 1).

> They kind of get. . . Well, I know they are professionals. They are doctors and nurses; they put on gloves and everything, but it is like they hardly want to touch you (Participant 9).

Although in most cases, they do not define this type of treatment as intentional transphobic encounters, the emotional impact that this type of adverse situation sometimes generates may contribute to avoiding seeking new health encounters later on for fear of suffering similar situations of discomfort or even rejection.

> It's hard for them to treat you with the correct pronoun, but from there to transphobia. . . well, no (Participant 1).

> Emotionally, it ends up shocking you. I mean, there comes a time when you say, "I don't want to go to the doctor because I feel uncomfortable, I have to lower myself to be treated. . .," and I do not go (Participant 13).

Several barriers or debilitating factors that transgender people have to deal with in relation to the access and use of health systems have been identified. The model of care focused on psychiatric diagnoses and psychological evaluations is described as pathologizing, discriminatory, and stigmatizing, with little relevance to the current health system.

> A good amount of psychologists and psychiatrists understand it as a pathology. We are not pathological, and we do not have a pathology. We are people as healthy as everyone else. Thus, many doctors use the element that is called clinical medicine that needs a diagnosis, and of course, our diagnosis is not clinical (Participant 12).

One of the first difficulties they face begins upon their arrival at the health center. People who have been unable (or unwilling) to change their name assigned at birth on the individual health card (IHC) usually ask the administration/admissions staff to change it to their given name, explaining the reason for the mismatch between their sex and gender. Although in some centers, they tend to write it in parentheses without any inconvenience, in most cases this occurs in vain, since either the health staff ignores their chosen name or they get confused and end up using both names.

> At the beginning, my health card did not have my name changed to a feminine form. So people, not seeing that there was no feminine name, treated me as masculine. In other words, it does not matter if they see you as female, but if there is a male name on the paper, they treat you as male. Until that changed, until I changed my name in Social Security on

the health card, and from then on, everyone knew I was a transgender girl and the treatment changed (Participant 1).

The murmurs, the stares of strangers, or even the discomfort in the reactions that occur around the transgender person when they are mentioned, produce an unpleasant and hostile effect, which is maintained once they enter the consultation with the health professional. As a result of previous bad experiences such as discriminatory situations in which they have even been annulled as persons, the interviewees reported always having a feeling of uncertainty, as they could not predict the reaction of the health professional when they mentioned their transgender identity.

It is not fear, but uncertainty in case I meet professionals who are not like that (trans-friendly) (Participant 7).

Although in some cases, they are grateful for the interest shown by health personnel in transgender issues, there is a certain degree of discomfort about having to give explanations or answers about their condition when the reason for the consultation is completely unrelated and distant from the topic. Discussing their genitalia or sex reassignment surgeries with a health professional who is not sensitized to the subject without apparent need is an uncomfortable exercise that is experienced as a violation of privacy. Some participants described it as morbid for the other person and out of context in many situations.

There was a case of another nurse that I had, and the subject came up while she was pricking me, why I needed this, and she said: "oh shit, they will operate on your genitals. . .," and I told her that it was not so easy and a very broad and private subject. She did not mean it in a bad way, but I found it a bit out of place (Participant 14).

Another present obstacle is the difficulty in the therapeutic relationship as a consequence of health professionals' lack of knowledge. In these cases, paternalistic attitudes typical of the biomedical model tend to appear in which decision-making is not shared because of the role of power and determination acquired by the health professional. The initial assessment of the patient may be affected if correct anamnesis and respectful physical examination are not performed. Some situations experienced during physical examinations generate a decrease in self-esteem, especially in those who do not feel completely comfortable when they have to expose their body in front of other people.

They have to put on gloves, but they do it with disgust. They put on gloves, put on their masks, and they are doing it with disgust, and you think they do not realize it, but they do realize it, and we have to put up with it because they are doing their job (Participant 9).

The looks, looks of strangeness, at the level of the body, right? When it's your turn to get undressed to have some part of your body looked at, those weird looks like. . . It's that feeling, of looking at a freak. . . It's very subtle, maybe the person is trained and everything, in many things, but they keep looking at you like: "How weird it is. . . What is this?" And it may be curiosity, it may be whatever, but it's super uncomfortable for me, because they are situations that one cannot avoid, I cannot avoid going to the doctor. If my back hurts, I cannot avoid going, and maybe I have to take off my shirt to be looked at, right? This is extremely uncomfortable (Participant 8).

## Training queries

The negative situations experienced by transgender people are often due to the poor training and lack of awareness of health professionals on transgender issues, which translates into aggressive ignorance toward them. The paternalistic attitudes, the disinterest of some professionals towards their demands, or the fear of psychological evaluations stand out as negative.

> First, I had to be evaluated by a psychologist, everything I wrote, because it was not him who had a conversation with me; it was me with a pen filling it in: "yes or no, yes or no. . ." but sheets, sheets, sheets, and sheets. And of course, I was distressed. . . so I did not feel that it helped me in any way (Participant 10).

The absence or insufficiency of these competencies compromises the health of the transgender population and interferes with the therapeutic relationship. Erroneous and/or late referrals to specialists or the feeling of having little control over the effects of hormones are some of the examples that the participants say they have experienced.

> It also suggests that neither in medical school, nor in nursing school, nor in psychology, in practically no career is anything talked about the subject of transsexuality; they do not know anything, so how can they know? (Participant 3).

> I do not think so, basically because there is a lot of ignorance nowadays, for example, without going any further, a hormonal clitoris, there is very little knowledge. And there are check-ups that you get that it's like: "oh shit, what happened here?", I mean: "I'm not used to a hormonal clitoris", and in this I think there is not a lot of knowledge (Participant 13).

This leads to situations of mistrust in the relationship with the professional or uncomfortable feelings of having to constantly give explanations. This is why the interviewees prefer to be asked directly, feeling part of the individualized care process and avoiding unilateral decisions that are probably erroneous.

> If you do not know, ask. I am not offended by being asked, so I prefer to be asked (Participant 6).

> There may be transgender guys who are hospitalized and someone comes to them with the bottle because they do not know anything and the guy who says: "hey, you better bring me the wedge because I am a transgender guy and the wedge suits me better because I have a pussy" No, I would always ask: "you told me you have pee, can I bring you the wedge?" And the person will already tell you: "no, I prefer the bottle" Of course, maybe you see a girl like me and you say: "Bring me the bottle," so you say: "The bottle? But well, you bring it and you will see what they say. Yes, and in the case that you haven't asked, you have brought it and they correct you and so on, don't act surprised: "oh! I didn't know, oh!" do not make it dramatic, take it discreetly, and that's it. So I bring you the wedge and that's it. . . go with a good attitude and do not take things for granted, ask questions with a predisposed attitude (Participant 3).

## Coping strategies

Despite this lack of competence on the part of the professionals, the participants accepted and justified their lack of knowledge by attributing it to unintentional errors due to inexperience and/or ignorance rather than intentional actions.

No, I would not say that there is any intention, at least in my case. I think it's more ignorance of not usually encountering these cases, and that they have not wanted, well, to investigate or. . . (Participant 13).

The need to overcome barriers and negative experiences leads transgender people to develop their own coping strategies. In general terms, the naturalization of the healthcare process is identified as a key action to face in healthcare encounters, as transgender people themselves are the ones who initiate an interaction that generates a climate of trust and normalization during the encounter. Some of the participants described this as "not attracting attention (Participant 1)" with the intention of going unnoticed or even avoiding generating positive discrimination, since they do not intend to be treated in a special or different way from other healthcare users.

[Doctors] have not been very incisive on the subject, such as: Are you transgender? Ah, well, very well, and naturally. Normalizing it. That is what you want in the end. You do not want to be treated as special. It's new for them, but for me it's not a big deal (Participant 7).

Some transgender people use a different strategy based on enhancing passing, a term attributed as an aid to adapt their gender identity to their felt gender, thus fitting into the binary social norms and giving them "protection" and "invisibility" in the eyes of the population. In this way, they can avoid mentioning their transgender identity in cases in which it is not medically relevant.

As I speak with my voice, I have not changed it, nor do hormones change your voice, so in that sense, it was shocking for people to see a "feminine" person speaking with their own voice. In this sense, people were shocked. It cannot be, a woman has to speak in a very feminine manner. What happens is that people are not educated, and since they are not educated, everyone reacts according to what they have experienced (Participant 1).

I suppose that there has been some progress. The more masculine you look, the more masculine you are treated. And that always. . . It's not something that bothers me. . . Well, in the beginning, it did bother me a lot, but psychologically, I was very sensitive, and obviously, that affected me (Participant 7).

Sometimes, they even describe occasions in which they themselves educate health personnel about transgender issues directly during visits. This exercise is carried out when professionals show a predisposition for this learning, as well as a receptive attitude to be trained by the users themselves, and it occurs among transgender people who present their own management and control over their health. This empowerment, generated by previous experiences, allows them to confront and stand up to health professionals in situations in which they are forced to claim and fight for their rights.

When I performed the tests for my surgery, of course, as I have this activist format, it got messy. I went to the anesthesiologist, and she told me, Do you know that you will be mutilated here? And I said, Excuse me? No, listen, you are very wrong. From the beginning, this approach should no longer be allowed because this question that you have asked me is 100% transphobic and goes against what Law 11/2014 says that the transgender issue is not any pathology, and you are considering it as a pathology. We are not mutilated, ma'am [. . .] And I told her: if you have to make a report so that they do not operate on me, do not worry, we will make a fuss (Participant 12).

## Discussion

Transgender people mostly perceive their experiences in the healthcare environment negatively, highlighting that on too many occasions they face traumatic care experiences when accessing healthcare services, both by healthcare and non-healthcare professionals. These findings are in line with those observed in the international literature in recent years [29–33].

Health-care disparities stemming from sexual orientation and healthcare encounters, attitudes, and beliefs impact the quality of care and health outcomes; these encounters in healthcare settings are frequently described by participants as dehumanizing and/or stigmatizing and may hinder their subsequent use of healthcare services due to a fear of discomfort or rejection [29, 33]. In line with the results of this study, the review by Zeeman et al. concluded that LGBTIQ+ people are more likely to experience health inequalities, although more studies are necessary to test the effect of these inequities on the health of this group, especially in the transgender and intersex population [29, 34]. Similarly, Lavorgna et al. evidence the number of health center changes is associated with center friendliness and the presence of discriminatory behaviors; thus, these attitudes affect LGBT health services use and health status [35].

Along the same lines, the qualitative research of the European Union's Health4LGBTI project evidenced the inequalities and barriers experienced by the LGBTI population, with special emphasis on transgender, intersex, and bisexual people [30]. Literature has shown LGBT patients fear being stigmatized or discriminated against; specifically, their previous negative experiences make them feel they are not receiving appropriate care from health providers, and this may lead to a delay in seeking medical care [36].

A large number of the study participants agreed that the current healthcare model, that is centered around psychological evaluations and psychiatric diagnoses in the care of the transgender population, encourages the generation of a pathologizing, discriminatory, and stigmatizing dynamic, which is a barrier to the healthcare of this group. In relation to this dynamic of exclusion, and in accordance with the requests of the study participants, in recent years, various international organizations have proposed new theoretical frameworks to address this problem from the perspective of human rights, public health, and mental health: the work of Agius and Tobler [37], or more recent documents focused on the field of health, such as the SESPAS report or the review by the same author in the Public Health Reviews [38, 39] stand out here.

In addition, in recent years, the international scientific community has been advancing along these lines, and the depathologization of transgender identity, in the latest edition of the International Classification of Diseases of the World Health Organization [40], which will come into force in 2022, is one of the great historical advances in this regard. At the Spanish level, a bill was presented in 2018 to protect the rights of transgender people and equalize the laws specific to each autonomous community, but the bill has not yet been approved and law 3/2007 of March 15 is still in force.

Thus, in relation to negative experiences with the health system, transgender people assert different demands that help to improve the care they receive; on the one hand, the protocolization of care for the transgender population is proposed to avoid the current pathologizing model and thus improve access to treatment. These demands correspond to the results obtained in the study by Lampalzer et al., conducted on LGBTI people, from which the terms depathologization, sensitization, inclusion, and awareness emerged [31]. On the other hand, emotional accompaniment and psychological therapies are requested as resources from the public health services portfolio, replacing current psychiatric evaluations.

It has been demonstrated that LGBT persons experience unreasonably higher burdens of physical and psychosocial health disorders [35]. Evidence also shows transgender people suffer

from higher rates of HIV and sexually transmitted diseases, cruelty, victimization, and mental health issues [36, 41]. Remarkably, 41% of transgender individuals have attempted suicide [42]. Several studies have pointed out this group's specific needs in the field of mental health, their greater susceptibility to developing mental health problems or emotional distress, and the difficulties they face in accessing specialized care [36, 43, 44]. Furthermore, in relation to accessibility in Catalonia, although medical care and surgical, psychological, and social treatment are included in the public portfolio [45], waiting lists for these services are often long and many people must seek alternatives, such as opting to have sex reassignment surgeries in other countries with reduced prices like Thailand, India, and Malaysia [46–48].

Likewise, the legal difficulties and requirements for changing one's name and gender among the transgender population are a manifestation of the problems faced by this collective in all areas, including healthcare [49, 50]. Several interviewees mentioned that this was an additional difficulty they faced during visits to primary care centers, as many professionals did not know how whether address them by their birth name or their chosen name. This is another example of a lack of training in this area among healthcare professionals. The FELGTBI + report [3] showed that 33% of the transgender people surveyed stated that the treatment provided by healthcare personnel regarding their gender identity had never or almost never been adequate. This problem is repeated at the international level, and there are a variety of documents on good practices and patient-centered care for this group (combined under the concept of affirmative care), such as the TRANSCARE approach of the University of Iowa or the recommendations of the CDC or WPATH [51–53].

Further, the transgender people interviewed call for improved training and sensitization of professionals in this field. Currently, they still encounter paternalistic attitudes typical of the biomedical model and a system in which the role of power lies with the professional, and shared decision-making and respect for individuality are not encouraged. The FELGTBI+ [3] study asked the transgender people surveyed to describe the level of health knowledge about the transgender reality they believed healthcare professionals possessed, and 75.3% expressed a low or very low level. The issue of the lack of training of healthcare professionals is a recurring theme in multiple studies [29–32, 41] and is one of the main points for improvement identified to reduce the barriers experienced by this collective in their access to healthcare services. Likewise, different authors advocate the inclusion of specific training on the health of transgender people, as well as on the LGBTI population in general, in the educational system, with special emphasis on the curricula of university health sciences degrees [16].

Improvements in the training of health professionals at the university level should be linked to the review of issues related to the LGBTIQA+ community, aspects of sex education, gender identity, and assessment of the needs of groups at greater risk of suffering discrimination, as is the case for transgender people [16, 52]. In this manner, universities could be spaces in which to acquire a critical view of the social and normative context that questions concepts as deeply rooted as heteronormativity and the pathologization of all identities that do not comply with established social norms, in addition to condemning transphobic attitudes that, unfortunately, are too often repeated in the healthcare environment [29, 31, 33, 34, 39].

Finally, and directly related to the last point, in consideration of these difficulties in the healthcare setting, the population studied has been forced to develop coping and empowerment strategies to overcome existing barriers. These include the naturalization of the healthcare process, adaptation to binary social norms (passing), and education of the healthcare community. The concept of passing derives from sociology, and, in the case of the transgender population, it is defined as the recognition of the transgender person with their own gender identity, and not with the gender they are assigned at birth [54–56]. Although this term has arisen in reference to a coping strategy used by the transgender community, it is another

example of the transphobia that exists at the social level and often contributes to perpetuating the discrimination and stigma suffered by this group. Similarly, other studies [34, 57, 58] mention some of these elements as effective individual strategies developed by this population in the face of the stressful situations and discrimination they face. Together with others, at the collective, educational, and healthcare levels, they can serve as a guide for future interventions to improve the transgender population's access to and use of health services and to enhance their empowerment and coping strategies in the face of stigma and inequality.

However, it is important to emphasize the essential importance of education to avoid inequalities derived from the binary conception of gender and heteronormativity and the importance of recognizing non-binarism, together with the depathologization already mentioned [29, 30, 37–39], with the aim of achieving a non-judgmental, inclusive, and respectful health and social context.

This study has several limitations that should be considered. First, it may be possible that some transgender persons who were interested in taking part in the study did not do so for fear that their participation could have a direct or indirect negative impact on the subsequent care they received from their healthcare professionals. To avoid this situation, the research team clarified that the confidentiality of the data was ensured throughout the research process and that the personal data of the participants would be modified to guarantee anonymity. The type of sampling used could have resulted in a sample of activist participants, and the data obtained may not reflect all transgender realities. Even so, an attempt was made to diversify the inclusion of participants as much as possible to reduce this limitation. Finally, the data obtained cannot be generalized, but they can be transferred to similar sociocultural contexts, which is the main characteristic of the methodology used.

## Conclusions

This study provides evidence of the needs and dissatisfaction of the transgender population in the healthcare setting. The inequalities faced by this population during the access and use of the healthcare system cause negative experiences during their interactions with professionals, generating discomfort during encounters. Despite the coping strategies developed by transgender people, such as the naturalization of the therapeutic process or passing, through which they aim to overcome the inequality and discrimination they may experience, it is necessary to make important changes in the healthcare provided to this population. The healthcare approach to transgender people must consider their specific needs, so strategies must be adopted to provide this group with quality, individualized, holistic, and respectful healthcare.

The depathologization of transgender identity would represent an advance and improvement in access to different treatments since it would eliminate the associated psychiatric diagnostic categories in addition to the legal procedures and requirements directly dependent on the diagnosis. Thus, in terms of inclusion, the free recognition of gender identity would allow non-sexist male-female stereotyping in a dichotomous manner, allowing the transgender population to become socially visible without prejudice.

Training in sexual and gender diversity must be included in the curricula of university courses in the health sciences. Healthcare professionals' lack of knowledge and awareness of transgender issues is one of the main barriers encountered by the transgender population. Therefore, further research, the development of protocols and care guides, and the training of health personnel will professionals better understand the reality of transgender persons and adopt specific interventions to modify and improve the care provided to this group.

## Acknowledgments

We would like to thank all the participants who shared with us their stories and time. We would also like to thank the associations Generem, Stop Sida, Transforma, Trànsit, Acathi, gTt-VIH, Àmbit, Grup d'Amics Gais-Lesbianes-Transexuals i Bisexuals de Barcelona (GAG) and Mrs. Maria Garro (RN, MSc) for their crucial help during the recruiting process.

## Author Contributions

**Conceptualization:** Mariela Aguayo-González.

**Data curation:** Mariela Aguayo-González.

**Formal analysis:** Mariela Aguayo-González.

**Methodology:** Kevin Santander-Morillas, Juan M. Leyva-Moral.

**Project administration:** Juan M. Leyva-Moral.

**Supervision:** Juan M. Leyva-Moral.

**Validation:** Mariela Aguayo-González.

**Visualization:** Kevin Santander-Morillas, Juan M. Leyva-Moral.

**Writing – original draft:** Kevin Santander-Morillas, Juan M. Leyva-Moral, Mariela Aguayo-González, Rebeca Gómez-Ibáñez.

**Writing – review & editing:** Kevin Santander-Morillas, Juan M. Leyva-Moral, Marta Villar-Salgueiro, Mariela Aguayo-González, David Téllez-Velasco, Nina Granel-Giménez, Rebeca Gómez-Ibáñez.

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
