## [Decision Letter · Decision Letter 0]

10 Jun 2022

PONE-D-22-09860TRANSALUD: A Qualitative Study of the Healthcare Experiences of Transgender People in Barcelona (Spain)PLOS ONE

Thank you for submitting your manuscript to PLOS ONE. After careful consideration, we feel that it has merit but does not fully meet PLOS ONE’s publication criteria as it currently stands. Therefore, we invite you to submit a revised version of the manuscript that addresses the points raised during the review process.

We look forward to receiving your revised manuscript.

Kind regards,

Luigi Lavorgna

Academic Editor

PLOS ONE

Journal Requirements:

4. Please include your tables as part of your main manuscript and remove the individual files. Please note that supplementary tables (should remain/ be uploaded) as separate "supporting information" files.

5. Please include a copy of Table 2 which you refer to in your text on page 9.

Reviewers' comments:

Reviewer's Responses to Questions

**Comments to the Author**

1. Is the manuscript technically sound, and do the data support the conclusions?

Reviewer #1: Yes

2. Has the statistical analysis been performed appropriately and rigorously? 

Reviewer #1: N/A

3. Have the authors made all data underlying the findings in their manuscript fully available?

Reviewer #1: Yes

4. Is the manuscript presented in an intelligible fashion and written in standard English?

Reviewer #1: Yes

5. Review Comments to the Author

Reviewer #1: In this study, Santander Morillas et al. conducted a descriptive analysis of issues transgender people face when accessing healthcare resources in Spain. Authors categorised issues in three axes: overcoming obstacles, training queries, and coping strategies. This study could help research staff and scientists to further stress the importance of specific training at University.

• The introduction is quite long and sometimes goes out of the scope of the present analysis. I would suggest to summarise this paragraph.

• Why you hide the ethical committee if I have the list of authors in the first manuscript page? Please describe the ethical committee

• Could you please introduce a reference for the following statement: “The data were analyzed descriptively and thematically following the method proposed by Colaizzi with the help of the Atlas.ti8 software.”?

• It would be useful to discuss the possible Health-care disparities stemming from sexual orientation according to disorders. Several papers have already described this topic and you could use them to discuss the possible different approach in different disciplines (see doi:10.1016/j.msard.2017.02.001; doi: 10.1016/j.cden.2021.06.007; doi: 10.1161/CIR.0000000000001003; doi: 10.1016/j.jaad.2018.02.042)

The aim is quite ambitious, as the amount of manuscripts on the topic has steeply surged within the last year in relation with both healthy conditions and diseases. I would better highlight why authors decided to only include manuscript published between January and February 2021 and also I would further introduce the possible association between the physical exercise and quality of life, including the sleeping status, in both healthy controls and in patients with other diseases (see this works Durucan et al. 2022 [10.1097/MRR.0000000000000519]; Cunha et al. 2021 [10.1016/j.numecd.2021.12.019]; Carotenuto et al. 2021 [10.3390/jcm10061234]; Beydoun et al. 2022 [10.1093/gerona/glac028]; Bruno et al. 2022 [10.1016/j.sleep.2022.01.002]). All the aforementioned manuscript usually referred to people’s condition during the period you choose for selecting manuscript and hence, during the more severe COVID 19 restrictions.

6. PLOS authors have the option to publish the peer review history of their article (what does this mean?). If published, this will include your full peer review and any attached files.

Reviewer #1: No

---

## [Author Response · Author response to Decision Letter 0]

30 Jun 2022

Thank you for your valuable comments. Please see the attached document with our changes.

---

## [Editor Report · Decision Letter 1]

5 Jul 2022

TRANSALUD: A Qualitative Study of the Healthcare Experiences of Transgender People in Barcelona (Spain)

PONE-D-22-09860R1

We’re pleased to inform you that your manuscript has been judged scientifically suitable for publication and will be formally accepted for publication once it meets all outstanding technical requirements.

Kind regards,

Luigi Lavorgna

Academic Editor

PLOS ONE
---

## [Editor Report · Acceptance letter]

7 Jul 2022

PONE-D-22-09860R1 

TRANSALUD: A qualitative study of the healthcare experiences of transgender people in Barcelona (Spain) 

Dear Dr. Leyva-Moral:

I'm pleased to inform you that your manuscript has been deemed suitable for publication in PLOS ONE. Congratulations! Your manuscript is now with our production department. 

Kind regards, 

on behalf of

Dr. Luigi Lavorgna 

Academic Editor

PLOS ONE